# Evaluation of the Antiviral Activity of Tabamide A and Its Structural Derivatives against Influenza Virus

**DOI:** 10.3390/ijms242417296

**Published:** 2023-12-09

**Authors:** Soo Yong Shin, Joo Hee Lee, Jin Woo Kim, Wonkyun Ronny Im, Kongara Damodar, Hyung Ryeol Woo, Won-Keun Kim, Jeong Tae Lee, Sung Ho Jeon

**Affiliations:** 1Department of Life Science and Multidisciplinary Genome Institute, Hallym University, Chuncheon 24252, Republic of Korea; tndyd8803@gmail.com (S.Y.S.); wngml9711@naver.com (J.H.L.); ultiway88@gmail.com (J.W.K.); wkronny.im@hallym.ac.kr (W.R.I.); 2Department of Chemistry and Institute of Applied Chemistry, Hallym University, Chuncheon 24252, Republic of Korea; kongaradamu@gmail.com (K.D.); whl1220@naver.com (H.R.W.); 3Department of Microbiology and Institute of Medical Science, College of Medicine, Hallym University, Chuncheon 24252, Republic of Korea; wkkim1061@hallym.ac.kr

**Keywords:** influenza viruses, antiviral drugs, tabamide A, structural derivatives, viral RNA synthesis

## Abstract

Influenza viruses cause severe endemic respiratory infections in both humans and animals worldwide. The emergence of drug-resistant viral strains requires the development of new influenza therapeutics. Tabamide A (TA0), a phenolic compound isolated from tobacco leaves, is known to have antiviral activity. We investigated whether synthetic TA0 and its derivatives exhibit anti-influenza virus activity. Analysis of structure–activity relationship revealed that two hydroxyl groups and a double bond between C7 and C8 in TA0 are crucial for maintaining its antiviral action. Among its derivatives, TA25 showed seven-fold higher activity than TA0. Administration of TA0 or TA25 effectively increased survival rate and reduced weight loss of virus-infected mice. TA25 appears to act early in the viral infection cycle by inhibiting viral mRNA synthesis on the template-negative strand. Thus, the anti-influenza virus activity of TA0 can be expanded by application of its synthetic derivatives, which may aid in the development of novel antiviral therapeutics.

## 1. Introduction

Influenza viruses comprise a major class of human respiratory pathogens, responsible for causing morbidity and mortality worldwide. They are the main cause of acute respiratory disease, often spreading worldwide owing to their high infectivity and antigen variability [1]. In particular, the influenza A virus (IAV) is constantly monitored because it infects many species and is one of the fastest-evolving family of viruses. Inside the host, the viruses inhibit the synthesis of cellular proteins and facilitate expression of their own proteins for viral transcription and replication. During infection, viral proteins interact with host proteins and exploit a variety of cellular pathways for their own benefit [2,3,4,5]. Infected cell pathways are hijacked by an array of intracellular signaling cascades, such as the phosphatidylinositol 4,5-bisphosphate (PI3K)/AKT pathway, which underlies the clinical manifestation of different stages of infection by viruses such as the Epstein–Barr virus, hepatitis B and C viruses, human immunodeficiency virus (HIV), and influenza virus [6]. Nonstructural protein 1 (NS1) of IAV interacts with various host cell proteins to suppress immune responses and promote viral replication [7]. NS1 stimulates the activation of the PI3K/AKT signaling pathway by directly binding to the p85β regulatory subunit of PI3K, a cellular lipid-metabolizing enzyme. Activation of the intracellular PI3K/AKT pathway not only controls the early stages of viral entry, but also inhibits early apoptosis at the late stages of infection and plays an important role in viral RNA expression and viral ribonucleoprotein complex (vRNP) localization [4,6,8]. Therefore, the interaction of the IAV NS1 protein with the host cell PI3K-AKT complex may promote viral replication by inhibiting host cell apoptosis [9]. 

The currently available anti-influenza therapeutic agents include matrix protein 2 (M2) ion channel and neuraminidase (NA) inhibitors. M2 inhibitors inhibit viral replication by binding to the M2, an integrated membrane protein responsible for the release of vRNP during early infection [10]. However, viral resistance to M2 inhibitors has been observed in several influenza A virus strains. NA activity is required to release new virions from infected cells. NA inhibitors bind to the active site of the NA protein and inhibit viral release from the infected host cells. The determination of the three-dimensional crystal structure and NA catalytic sites led to the design of the first two potent inhibitors, zanamivir and oseltamivir (Tamiflu), which were approved for clinical use [11,12]. However, the need to develop new antiviral agents has emerged because of the side effects of existing antiviral drugs and the emergence of resistant influenza virus strains. Accordingly, research on the development of antiviral agents with fewer side effects from medicinal plants has been in focus in recent times [13,14].

Phenolic compounds, the secondary metabolites of plants, are derivatives of the pentose phosphate, shikimate, and phenylpropanoid pathways, and have considerable physiological and morphological importance. They are the most common bioactive compounds in plants, fungi, marine algae, and other natural sources, and exhibit biological properties that are known to benefit human health and affect the plant as well. Certain groups of phenolic amides have attracted considerable attention for their interesting pharmacological functions, such as anti-inflammatory, antioxidant, anticancer, and anticoagulant activities [15,16,17]. Currently, more than 8000 different types of plant phenolics are known and being studied for use as functional foods or cosmetic additives because of their natural origins and low toxicity. 

Tabamide A (TA0) is a natural phenolic amide isolated from the leaves of *Nicotiana tabacum*, an herbaceous plant grown worldwide as a major commercial source of tobacco. In addition to tobacco, which is used as a drug for its mild arousal effect, *N. tabacum* is used in folk medicine as an insecticide, sedative, and analgesic [18,19]. Several phenolic compounds isolated from *N. tabacum* exhibit anti-tobacco mosaic virus (anti-TMV) and anti-HIV activity [20,21]. Expecting potent antiviral and other beneficial activities, TA0 and its derivatives were synthesized and their biological functions were investigated. TA0 effectively suppressed nitric oxide (NO) production, an indicator of anti-inflammatory activity [22]. More importantly, we recently found that TA0 also exhibits anti-influenza activity, in addition to anti-TMV activity. 

In this study, synthetic TA0 and its derivatives were used to elucidate the structure–activity relationship of these compounds. In addition, novel compounds with improved anti-influenza activity were designed and synthesized, and their efficacies were evaluated.

## 2. Results

### 2.1. Anti-Influenza Activity of Synthetic TA0 and Its Derivatives

Previously, we synthesized the natural products tabamides A-C and their derivatives [20,22], of which, only tabamide A (termed TA0 here) exhibited anti-influenza activity in preliminary experiments. As shown in Figure 1A, treatment of Madin–Darby canine kidney (MDCK) cells with TA0 protected them from the cytopathic effects caused by viral infection. The viability of virus-infected cells also increased by TA0 treatment in a dose-dependent manner. The expression levels of the viral *M2* transcript and NS1 protein were also significantly reduced under the same conditions (Figure 1B,C). 

Based on these results, primary derivatives (TA11–TA15) of the TA0 structure, modified at three positions (indicated with red boxed in Figure 2A), were synthesized to investigate whether these functional groups are vital for antiviral activity (Figure 2A). The inhibitory effects of all primary derivatives on viral infection were lower than those of TA0 (Figure 2B,C), indicating that antiviral efficacy was reduced when any of the three positions in the TA0 structure were altered. The hydroxyl (-OH) group at R’ and the double bond between C7 and C8 (red box) are essential for maintaining antiviral activity. This is indicated by the observation that the antiviral activity is completely lost by methylation of -OH at R’ (TA11) or reduction of the double bond (TA12 and TA13). Replacement of -OH with -H at R’ (TA14) reduced NS1 viral protein levels (Figure 2B), but this modification was unable to reduce viral titers (Figure 2C). 

In contrast, methylation of -OH at R (TA15) significantly reduced the viral titer, as in the case of the TA0 treatment, but partially suppressed NS1 expression. 

### 2.2. Synthesis of Secondary Derivatives Based on the Relationship between the Structure of Tabamide A and Its Antiviral Function 

Based on the first-round identification of the functional groups of TA0 that are important for maintaining antiviral activity, secondary derivatives (TA21–TA25) were designed and synthesized by modifying other functional groups available to improve antiviral activity (Figure 3). The altered locations are indicated by α and β (red boxes). TA21 and TA22 were designed by replacing the existing amide group with an ester or thioester at the α site. TA23 was synthesized by reducing the carbonyl group to a methylene group at the β site, and TA24 was prepared by changing benzofuran to benzothiophene in TA0. For TA25, the structural changes in TA23 and TA24 were combined to remove the carbonyl group at the β site and convert benzofuran to benzothiophene. All synthesized compounds were investigated for their cytotoxic effects expressed as 50% cytotoxicity concentration (CC_50_).

Among all the secondary derivatives of TA0, only TA22 (CC_50_ = 76.07 μM) exhibited significant effects on cell proliferation, even at the maximum concentration of 200 μM used in the experiment. Therefore, secondary derivatives, except TA22, can be considered safe for cells.

### 2.3. Secondary Derivatives of TA0 Effectively Inhibit IAV Infection

To test the inhibitory activity of TA0 and its secondary derivatives (TA21–TA25) against IAV infection, the expression levels of viral proteins and the number of plaques produced after infection were investigated (Figure 4). Expression of the NS1 protein decreased when treated with TA0 or its derivatives (Figure 4A). TA22 did not completely inhibit viral protein expression.

As shown in Figure 4B, the amount of virus produced in the infected cells was reduced by treatment with TA0 or secondary derivatives. Treatment with 100 μM TA0, TA21, or TA22 reduced the number of viral plaques to 11, 28, or 4%, respectively. TA23 treatment reduced the amount of virus by 0.3%. In particular, TA24 and TA25 showed much higher antiviral activity than TA0. These results suggested that replacing the amide group with an ester group (TA21) or a thioester group (TA22) at the α-site did not affect or reduce the antiviral effect, and removing the carbonyl group from the β-site (TA23) slightly enhanced the antiviral activity. The most noteworthy finding was the substitution of furan with thiophenes (TA24 and TA25). Viruses were not detected after treatment with TA24 and TA25; a combined change at the β-site and thiophene, showed the same result. Similar results were obtained when TA23, TA24, and TA25 were administered at low doses, and the total viral protein expression profile was examined (Figure 4C). 

To quantify the antiviral efficacy of the secondary derivatives of TA0, the concentration of the compound that inhibited viral activity by 50% (IC_50_) was measured. As shown in Figure 5, TA24 (IC_50_ = 1.3 μM) and TA25 (IC_50_ = 0.38 μM) had more efficient antiviral activity than TA0 (IC_50_ = 2.72 μM). IC_50_ of TA23 (5.4 μM) was higher than that of TA0, although treatment at higher concentrations (100 μM) more effectively reduced viral infection (Figure 5). Because the CC_50_ values of all compounds in the table exceeded the measured concentration of 200 μM, the selectivity index (SI) value representing the maximum antiviral activity with minimal cytotoxicity is indicated as > 200/IC_50_. In particular, TA25 showed an antiviral activity seven times higher than that of TA0. Based on the results of all in vitro experiments, it was concluded that TA25 is the most effective antiviral candidate among the TA0 derivatives. 

Next, we analyzed the antiviral activity of TA0 and TA25 in vivo using mice infected with the mouse-adapted influenza virus. To measure the antiviral activity, weight loss and mortality of the mice were monitored for 14 days after viral infection. Viral challenge with 2 units of a 50% lethal dose (LD_50_) resulted in the death of mice within 8 days (Figure 6A, IAV); however, all mice in the groups treated with 20 μL of 10 mM TA0 or TA25 survived, similar to the uninfected controls (mock). In addition, the body weight of mice treated with TA0 or TA25 decreased slightly, but immediately returned to normal, whereas in mice infected only with the virus, continuous weight loss was observed till death (Figure 6B). These results suggested that TA0 and its derivative TA25 clearly inhibited viral amplification both in vitro and in vivo.

### 2.4. Inhibitory Role of TA25 on Viral Infection and Amplification

NS1, an IAV viral protein, plays an important role in inducing the activation of the AKT- S6 kinase (S6K) signaling pathway for viral protein synthesis [6,23,24]. AKT activation enhances the phosphorylation of ribosomal protein S6, which is critical for viral protein synthesis. As shown in Figure 7A, AKT phosphorylation at residue T308 was associated with NS1 expression after viral infection. TA25 significantly reduced viral NS1 expression. In addition, TA25 completely inhibited the phosphorylation of AKT and S6. These results could be explained by inhibition by TA25 of the AKT or its upstream signaling pathways, thereby reducing the synthesis of NS1 and other viral proteins. Another explanation is that TA25 acts early in the viral life cycle to inhibit RNA synthesis from *NS1* gene. Our experiments, in which NS1 was overexpressed from an exogenous plasmid (pCAGGS; Addgene, Watertown, MA, USA) in the absence of viral infection, supported the latter hypothesis. TA25 also inhibited AKT-S6K signaling induced by NS1 overexpressed under control of the β-actin promoter of the plasmid (Figure 7A). These results suggest that TA25 possibly inhibits the NS1-induced AKT activation signaling pathway via a mechanism unrelated to viral infection. Similar results were obtained with TA0, the parent compound of TA25. Therefore, TA25 is expected to act at an early stage of viral gene expression, that is, during the synthesis of positive (+) strand RNA. The first synthesized (+) strand RNA (mRNA) is primarily transcribed from the viral negative (−) strand RNA (vRNA), and serves as a template for protein synthesis. In contrast, the (−) RNA strand has two origins. One is exogenous vRNA, and the other is synthesized anew from the (+) strand RNA (complementary RNA, cRNA) intermediate in the nucleus of host cells. We measured the levels of (+) or (−) stranded RNA in cells 1–8 h after viral infection. The amount of (+) strand RNA of NS1 produced 4 and 8 h after viral infection was significantly reduced by TA0 or TA25 treatment (Figure 7B, top).

Similar results were obtained when comparing the levels of (−) strand RNA, which is considered to be newly synthesized using cRNA as a template, 8 h after infection. However, the (−) strand RNA present at 4 h postinfection was most likely derived from exogenous viral particles, and no quantitative change was observed with TA0 or TA25 treatment (Figure 7B, bottom). These results suggest that TA0 and its derivative TA25 play an inhibitory role in RNA-dependent (+) strand RNA synthesis in the early stages of viral replication. 

## 3. Discussion

Phenolic compounds originating from plants with antiviral activity could be beneficial in terms of safety, with fewer side effects and several additive effects, such as anti-inflammatory and antioxidant effects. Viral infections sometimes trigger cytokine storms, owing to the activation of the host immune system. Therefore, the anti-inflammatory effects of natural phenolic compounds may downregulate these undesirable effects. TA0 did not show cytotoxicity, even when treated with a high dose (CC_50_ ≥ 200 μM). In addition to the inhibitory effect of TA0 on TMV, we showed that this compound acts against a wide range of viruses (manuscript in preparation). TA0 also reduced NO production in lipopolysaccharide-stimulated Raw 264.7 macrophages [22]. Therefore, TA0 and its structural derivatives are safe and suitable antiviral agents that can modulate innate immunity against viral infections.

As shown in Figure 5, the IC50 value of TA25 (IC50 = 0.38 μM) is comparable to commercial anti-IAV compounds such as Oseltamivir or Zanamivir. These compounds have different IC50 ranges depending on the subtype of virus and the analysis system. As neuraminidase inhibitors, the IC50s of Oseltamivir and Zanamivir against H1N1 strains were 0.92 nM and 1.34 nM, respectively [25]. According to the plaque reduction assay, the IC50 of Zanamivir was 0.19 μM for H1N1 and 15.93 μM for H3N2 [26]. In our experiments, the IC50 values of TA0 and TA25 against H3N2 were 3.1 and 0.4 μM, respectively, which is more sensitive than the reported value of Oseltamivir (IC50 = 4.9 μM) [27]. 

Flavonoids are the most common group of naturally occurring phenolic compounds that exhibit various antiviral activities [28,29]. Their mode of action against viral infection may be referenced in further studies on the antiviral activity of TA0 and its derivatives. Flavonoids act as inhibitors at different stages of viral infection in the free state, or as glycoside derivatives, which can increase the solubility of flavonoids. For example, quercetin inhibits viral entry by interacting with the hemagglutinin (HA) envelope protein, IAV [30]. Many flavonoids, including quercetin, have shown high binding affinity for the S protein of severe acute respiratory syndrome coronavirus 2 (SARS-CoV-2) [31]. However, quercetin also inhibits Ebola virus by targeting the VP24 protein in the cytoplasm, which suppresses the host’s antiviral immune system [32]. TA0 and TA25 did not prevent viral entry into host cells, as these molecules were unable to restore viral hemagglutination inhibition. Instead, the negative strand of the virus was detected in the nucleus, and its amount was not reduced by TA0 or TA25 treatment until 4 h after viral infection (Figure 7B). However, the (+) strand RNA encoding viral proteins was reduced by TA0 or TA25 treatment, even 1 h after infection. These results suggest the possibility that tabamide A and its derivatives act as RNA-dependent RNA polymerase (RdRp) inhibitors that interfere with viral mRNA synthesis from (−) strand genome.

Two influenza antiviral drugs, favipiravir and baloxavir marboxil, targeting the RNA-dependent RNA polymerase (RdRp) have been developed [33,34]. Favipiravir (T-705) is a precursor of favipiravir-ribofuranosyl-5′-triphosphate, which competes for the purine bases of RdRp, and is incorporated into viral RNA [35]. These nucleoside analogs induce mutations in the progeny RNA, consequently inhibiting viral replication. In contrast, *baloxavir* marboxil (Xofluza) inhibits the cap-dependent endonuclease activity of RNA polymerase acidic (PA) protein [34], and has been approved by the US Food and Drug Administration. Baloxavir acid, the active form produced by the hydrolysis of *baloxavir* marboxil, inhibits a process called cap snatching. Viral RdRp binds to the 5′-cap structure of the host transcript and cleaves it with endonuclease activity, and the resulting capped primer is used to initiate viral mRNA synthesis. Although the precise mechanism remains unknown, the synthesized compounds are expected to act during the early stage of viral RNA synthesis.

In conclusion, we have demonstrated the antiviral efficacy of TA0 and its derivatives using in vitro and in vivo animal model systems. The significance of our finding that TA25 specifically inhibits the synthesis of viral mRNA during the early stages of infection is that this molecule may be used in combination with previously known neuraminidase. In addition, this ability to inhibit RNA suggests the possibility of being a viral therapeutic candidate that is not limited to influenza viruses, but can be applied to a wide range of RNA viruses. Regarding this, positive results are being obtained for SARS-CoV-2 and Zika virus (paper in preparation), and additional research is underway to identify the exact target molecule and its exact mechanism of action.

## 4. Materials and Methods 

### 4.1. TA0 (Tabamide A) and Its Derivatives

TA0 and its derivatives (TA11–13) used in this study were synthesized using Stobbe condensation and amide coupling reactions, as previously reported (Appendix A) [22]. TA14 and TA15 were obtained in the presence of amide coupling reagent hexafluorophosphate azabenzotriazole tetramethyl uronium. Compared to TA0 synthesis, the amine part (**27**) (Appendix A), originated from simple acetophenone (for TA14) and 4-methoxybenzofuran acid (**17** for TA15), were utilized. Base-mediated esterification and 1-ethyl-3-(3-dimethylaminopropyl)carbodiimide/4-dimethylaminopyridine-mediated coupling produced TA21 and TA22, respectively. In TA23 synthesis, 4-hydroxyphenethylamine (HE) was employed as an amine component. TA24 and TA25 were accomplished by amide coupling protocol by using benzothiophene acid (**16**). TA24 and TA25 have structural similarity to TA0 and TA23, wherein benzofuran acid (**15**) was introduced as acid part, respectively. All the target compounds were established from their spectral (^1^H- and ^13^C-nuclear magnetic resonance (NMR) and mass spectrometry (MS)) data (see Appendix A).

### 4.2. Cell Culture

Madin–Darby canine kidney (MDCK) cells and human lung adenocarcinoma A549 cells were purchased from the Korean Cell Line Bank (Cat. No., 10034 and 10185, KCLB, Seoul, Republic of Korea). MDCK cells were maintained in minimum essential medium (MEM, HyClone, Logan, UT, USA) supplemented with 10% fetal bovine serum (FBS, GenDEPOT, Grand Island, NY, USA) and 1% antibiotic-antimycotic (Thermo Fisher Scientific, Waltham, MA. USA). A549 cells were cultivated in Dulbecco’s modified Eagle’s medium (DMEM) (Corning Life Science, Glendale, AZ, USA) supplemented with 10% FBS and 1% antibiotic-antimycotic. 

### 4.3. Virus and Viral Infection

Influenza virus A/Brisbane/59/2007 (H1N1) and A/Brisbane/10/2007 (H3N2) were obtained from the National Culture Collection for Pathogens (Cat. No., NCCP 42464, Osong, Republic of Korea). The mouse-adapted A/Brisbane/59/2007 strain was a kind gift from Prof. S.H. Park (Korea University, Seoul, Republic of Korea). MDCK cells were dispensed into T75 flasks (SPL Life Science, Pocheon, Republic of Korea) at 2 × 10^6^ cells/flask and cultured for 24 h. After washing the medium twice with Dulbecco’s phosphate-buffered saline (DPBS, Corning Life Science, Glendale, AZ, USA) without calcium and magnesium, cells were infected with virus in MEM supplemented with tosyl phenylalanyl chloromethyl ketone-treated trypsin (Sigma-Aldrich, St. Louis, MO, USA) for 1 h at 34 °C. Virus-infected cells were further incubated at 34 °C for 48 h after changing the medium. BALB/c mice (5 weeks old, male) were purchased from Daehan bio link (Eumsung, Republic of Korea) and maintained at an ABSL2 facility of Hallym University. Mice were respiratory anesthetized with 2% isoflurane (Hanapharm Co. Ltd., Seoul, Republic of Korea), and then challenged with mouse-adapted influenza virus at 0.5 or 2 units of 50% mouse lethal dose (LD_50_) through nasal route. After 1 h, 20 μL of 10 mM TA0 or its derivative was intranasally administered. Changes in body weight and mortality were evaluated daily for 14 days. The animal experiments were conducted in accordance with ethical guidelines approved by the Institutional Animal Care and Use Committee of Hallym University (HallymR1(2020-55)).

### 4.4. Determination of Virus-Induced Cytopathic Effects

MDCK cells were seeded in a 96-well plate at 1.2 × 10^4^ cells/well and cultured for 24 h, followed by virus infection at 1 MOI and TA0 treatment. After removing the supernatant, cells were treated with water soluble tetrazolium salt (WST)-1 (DoGenBio, Seoul, Republic of Korea) and reacted at 34 °C for 1 h. Cell viability was determined by measuring absorbance at 450 nm using a microplate reader (Thermo Fisher Scientific, Waltham, MA. USA). After virus infection, the cytopathic effect was observed with a JuLI FL microscope (NanoEntek, Seoul, Republic of Korea). 

### 4.5. Plaque Assay

Plaque forming units (PFUs) were measured to determine infectious virus titers. MDCK cells were dispensed into 24-well plates at 2 × 10^5^ cells/well and cultured for 24 h. The remaining medium was then removed and the virus was diluted 1/10-fold from 10^−2^ to 10^−7^ and infected at 100 μL per well. After infecting the cells, they were shaken every 15 min for 1 h to prevent drying of the cells. For the agarose overlay assay, the supernatant was removed and the cells were overlaid with medium containing 1% NuSieve GTG Agarose (Takara Tokyo, Japan). After incubation for 48 h, 4% paraformaldehyde solution (Biosesang, Seongnam, Korea) was dispensed into each well and the cells were fixed for 20 min. Then, the cells were stained with 0.2% crystal violet and the number of plaques formed on the plate was quantitatively measured.

### 4.6. Reverse Transcription and Real-Time PCR Analysis

Cells were treated with 500 μL TRIzol^®^ Reagent (Thermo Fisher Scientific, Waltham, MA. USA) and 100 μL chloroform (Sigma-Aldrich, St. Louis, MO, USA) to extract intracellular RNA. For the reverse transcriptase polymerase chain reaction (RT-PCR), cDNA was synthesized using amfiRivert cDNA Synthesis Platinum Master Mix (GenDEPOT, Katy, TX, USA), and then PCR was performed using AccuPower^®^ Taq PCR Premix (Bioneer, Deajeon, Republic of Korea). Primers used to synthesize first strand cDNA were oligo dT primer or negative (−) strand-specific primer for segment 8 of IAV. Quantitative real-time PCR (qRT-PCR) was analyzed using Exicycler™ 96 (BIONEER, Daejeon, Republic of Korea) equipment by mixing ExcelTaq 2X Q-PCR Master Mix (SMOBIO, Hsinchu, Taiwan). The forward and reverse primers for viral gene detection were M2; 5′-TGAGCCTTCTAACCGAGGTCGAAA-3′ and 5′-CCACAATATCAAGTGCACAATCCC-3′, NS1; 5′-TCGCGGTACCTAACTGACATGACT-3′ and 5′-CTGGTCCATTCTGACACAAAGAGG-3′. As a control, canine glyceraldehyde-3-phosphate dehydrogenase (GAPDH) gene was amplified using the following primers; 5′-AATTCCACGGCACAGTCAAGGC-3′ and 5′-AACATACTCAGCACCAGCATCACC-3′.

### 4.7. Immunoblot Analysis

Total cellular protein was extracted using PRO-PREP Protein Extraction Solution (iNtRON biotechnology, Daejeon, Republic of Korea). The supernatant containing the soluble protein was separated by size on a sodium dodecyl sulfate poly acrylamide gel (SDS-PAGE) and transferred to a PVDF membrane (Merck, Darmstadt, Germany). The membrane was blocked by incubation in 5% skim milk (BD Difco, Sparks, MD, USA) dissolved in 1X Tris-buffered saline with Tween-20 (TBS-T). After 1 h, the following primary antibodies obtained from Cell Signaling Technology (CST, Danvers, MA, USA) were added for binding to human proteins: rabbit anti-Akt, rabbit anti-Phospho-Akt (Thr308), rabbit anti-p70 S6 kinase, rabbit anti-Phospho-p70 S6 kinase, and mouse anti-β-actin. Viral proteins expressed in host cells after infection were detected using mouse anti-Influenza A ns1 (Santa Cruz, Bergheim, Germany) or rabbit anti-influenza A H1N1 (Novus Biologicals, Abingdon, UK). As a control, rabbit anti-GAPDH antibody (CST, Danvers, MA, USA) was used for detection of canine protein. After the primary antibody reaction, the secondary antibody, anti-rabbit or anti-mouse IgG-horseradish peroxidase (HRP, CST, Danvers, MA, USA), was reacted for 1 h, and then SuperSignal™ West Femto Maximum Sensitivity Substrate (Thermo Fisher Scientific, Waltham, MA. USA) reagent was applied in the same ratio. Detection of the desired target protein was confirmed using SUPERNOVA-Q1800 (CENTRONICS. Inc., Daejeon, Republic of Korea). 

### 4.8. Statistical Analysis

In all experimental data, the mean and standard deviation (SD) of the independent experimental group are shown as mean ± SD (*n* = 3) of three iterations. The statistical significance was compared by Student’s *t*-test method using the GraphPad Prism^®^ Version 5.0 (GraphPad Software, San Diego, CA, USA) program (ns; not significant, * *p* < 0.05, ** *p* < 0.01, *** *p* < 0.001, and ^###^
*p* < 0.001).

## 5. Conclusions

Structure–activity relationship analysis showed that two hydroxyl groups and the double bond between C7 and C8 are important for the anti-influenza activity of tabamide A (TA0). TA25, an enhanced structural form of TA0 for antiviral activity, effectively inhibits the synthesis of viral mRNA during the early viral infection cycle. These substances could be used in the development of novel antiviral therapeutics.

## Figures and Tables

**Figure 1 ijms-24-17296-f001:**
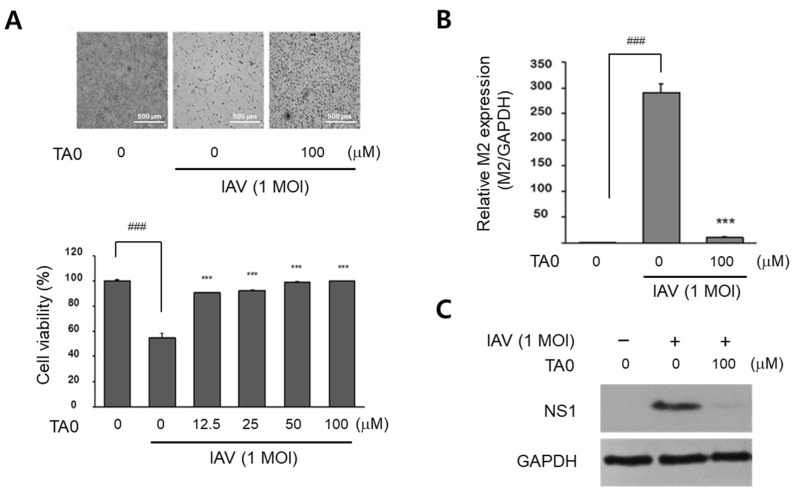
Tabamide A has antiviral activity against IAV infection. MDCK cells were infected with 1 MOI of IAV and treated with the indicated amount of TA0 for 24 h. (**A**) Virus-induced cytopathic effects were observed under a microscope (top), and cell viability was determined by measuring the absorbance at 450 nm (bottom). (**B**) Relative viral M1 RNA expression following viral infection and TA0 treatment was determined by qRT- PCR. (**C**) The expression level of NS1 protein was measured by immunoblot analysis. All values are expressed as mean ± SD (*n* = 3). ^###^
*p* < 0.001 compared to the cell control and *** *p* < 0.001 compared to the virus treated control. TA0: tabamide A; IAV: influenza A virus; MDCK: Madin–Darby canine kidney; MOI: multiplicity of infection; qRT-PCR: quantitative real-time polymerase chain reaction; NS1: nonstructural protein 1; GAPDH: glyceraldehyde-3-phosphate dehydrogenase; SD: standard deviation.

**Figure 2 ijms-24-17296-f002:**
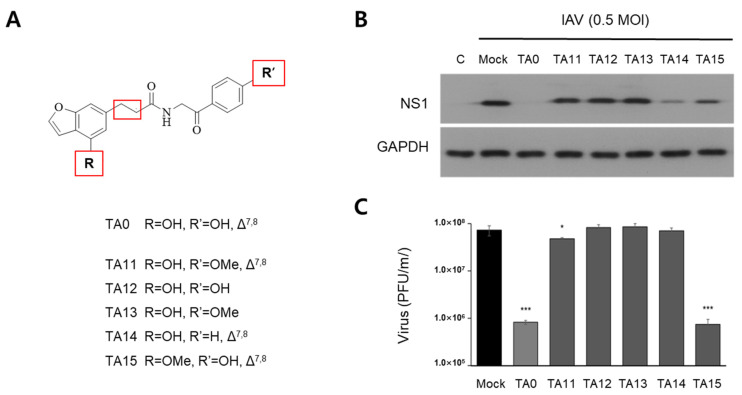
Evaluation of antiviral efficacy of primary structural derivatives of TA0. (**A**) Structures of TA0 and its structural derivatives (TA11-TA15). Δ^7,8^ indicates that there is a double bond between 7 and 8. (**B**,**C**) MDCK cells were infected with 0.5 MOI of virus and treated with 100 μM of TA0 or primary derivatives. After 24 h of treatment, changes in the expression level of NS1 protein were measured by immunoblot analysis (**B**). The number of plaques produced 48 h after infection with the serially diluted virus was determined quantitatively (**C**). Infectious virus titers were expressed as PFU/mL, and results are shown as mean ± SD (*n* = 3). * *p* < 0.05 and *** *p* < 0.001 compared to the virus treated control (mock). TA11–15: synthetic derivatives of TA0; PFU: plaque forming unit.

**Figure 3 ijms-24-17296-f003:**
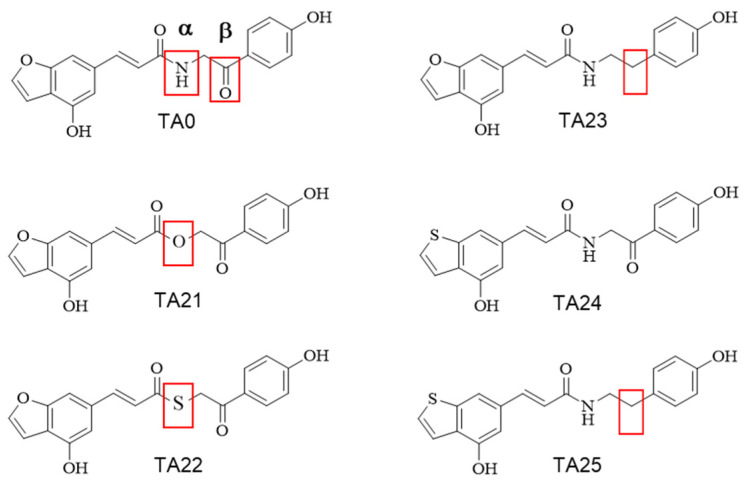
Structure of secondary derivatives of TA0. Secondary derivatives of TA0 were synthesized by substituting three positions of α and β, indicated by squares in the structural diagram of TA0.

**Figure 4 ijms-24-17296-f004:**
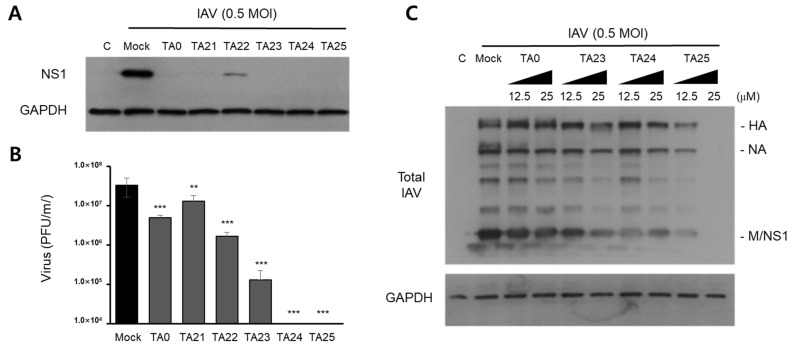
Secondary derivatives of TA0 have enhanced anti-influenza virus activity. (**A**) MDCK cells were infected with 0.5 MOI of IAV and treated with TA0 or its secondary derivatives at a concentration of 100 μM for 24 h. The expression level of NS1 protein was measured by immunoblot analysis. (**B**) The number of plaques produced by viral infection was quantitatively determined and expressed as PFU/mL. Results are expressed as mean ± SD (*n* = 3). (**C**) and IAV infection (mock), and ** *p* < 0.01 and *** *p* < 0.001 compared to the mock. (**C**) Expression levels of total IAV protein were examined by immunoblot analysis using an anti-IAV antibody after treatment with TA0 or its derivatives (TA23–TA25) at concentrations of 12.5 and 25 μM. TA23–25: synthetic derivatives of TA0.

**Figure 5 ijms-24-17296-f005:**
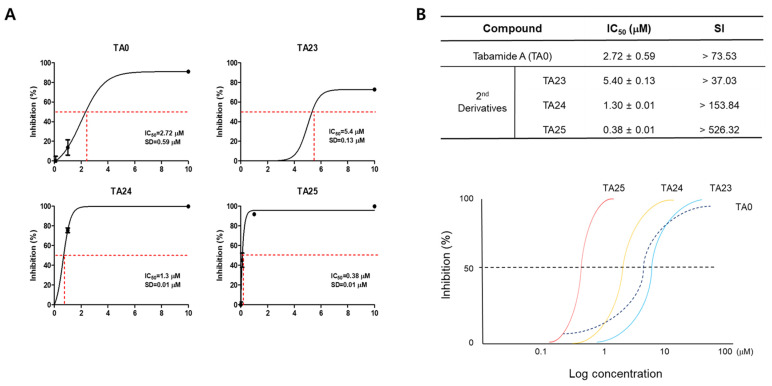
Half maximal inhibitory concentration (IC_50_) of TA0 and its derivatives. (**A**) IC_50_ values were determined by plaque inhibition assay. MDCK cells were infected with 0.01 MOI of virus and treated with various concentrations (0.1 to 100 μM) of TA0, TA23, TA24, and TA25. After 48 h, the number of plaques was measured. Black line: graph by actual data; red dot cross line: IC_50_ concentration (50% inhibition/concentration) (**B**) IC_50_ of TA0 and its derivatives are listed in table (top), and a hypothetical dose–response curve based on the measured values is shown in a graph for efficacy comparison between each compound (bottom). Results are presented as mean ± SD (*n* = 3). Selectivity index (SI) = CC_50_*/IC_50_. In the case of cytotoxicity concentration 50% (CC_50_) value, the maximum concentration of 200 μM or higher in the experimental group is not the measured value, but more than the maximum treatment concentration.

**Figure 6 ijms-24-17296-f006:**
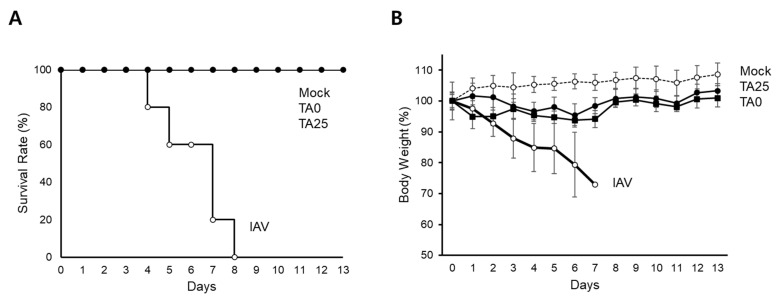
Tabamide A and its derivative TA25 inhibit influenza virus infection in vivo. Mice were challenged with 2 units of LD_50_ (**A**) or sublethal dose (**B**) of mouse-adapted influenza virus via the intranasal route. After 1 h, TA0 or TA25 were intranasally administered and survival rate or body weight loss were observed for 14 days. Results are shown as mean ± SD (*n* = 5 per group in the experiment). Mock: noninfected control; IAV: virus-infected control; LD: lethal dose.

**Figure 7 ijms-24-17296-f007:**
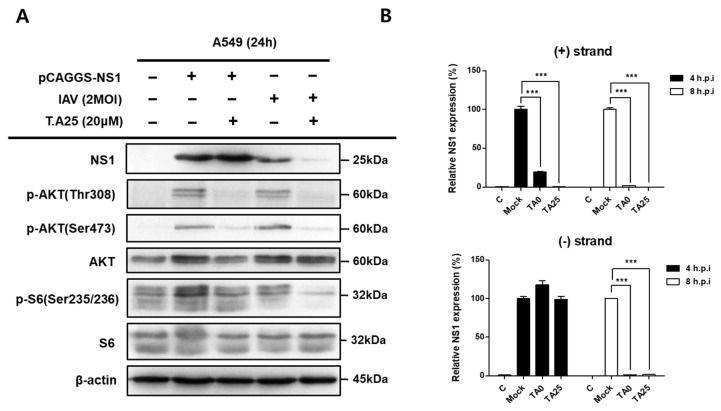
TA25 inhibits viral amplification in the early stages of viral infection. (**A**) A549 cells were infected with 2 MOI virus or transfected with a plasmid overexpressing the NS1 protein and treated with 20 μM TA25. The expression level of each protein was determined by immunoblot analysis using a specific anti-human protein antibody. (**B**) A549 cells were infected with 2 MOI virus for 1 h, and the medium containing virus was replaced and further cultured. After 4 and 8 h of infection, the relative expression of (+) strand RNA (mRNA and cRNA) or (−) strand RNA (vRNA) of the NS1 gene was measured by qRT-PCR. Results are presented as mean ± SD (*n* = 3). *** *p* < 0.001 compared to the virus treated control (mock).

## Data Availability

All data generated in this study are contained within the article or the Appendix A.

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
