# Peer review of "Evaluation of the Antiviral Activity of Tabamide A and Its Structural Derivatives against Influenza Virus"

_ijms, 2023, doi:10.3390/ijms242417296_

Round 1
Reviewer 1 Report
Comments and Suggestions for Authors
The manuscript titled "Evaluation of the Antiviral Activity of Tabamide A and Its Structural Derivatives Against Influenza Virus" Soo Yong Shin et al, presents a comprehensive study on the antiviral properties of Tabamide A (TA0), from tobacco leaves, and its synthetic derivatives. The study introduces new antiviral compounds derived from TA0, contributing to the development of potential therapeutics against influenza viruses. The detailed structure-activity relationship analysis shows how structural modifications of TA0 affect antiviral activity and provides valuable insights into designing more potent antiviral drugs. The efficacy of TA0 and its derivatives was demonstrated in vitro and in vivo (mouse models). The study investigates the mechanism of action of these compounds, finding that TA25 particularly inhibits the synthesis of viral mRNA in the early stages of infection. The compounds showed minimal cytotoxicity at effective antiviral concentrations, suggesting they are safe for therapeutic use.
In conclusion, the study offers significant insights into the development of novel antiviral agents derived from TA0 by targeting influenza A virus. In my view the study is sound enough to be published in its current form. However, I recommend minor revision.
Comments:
1. Please discuss the future directions/recommendations and limitations of this study in a separate section.
2. The study already provides insights into the mechanism of action, more detailed studies on how these compounds interact with viral components would be beneficial. For this purpose, in-silico docking, and simulation can be incorporated in this manuscript.
3. Studies on the potential development of viral resistance to these compounds would be a practical addition to understand their long-term effectiveness.
4. The inclusion of pharmacokinetic and pharmacodynamic data would provide more comprehensive information about the behavior of these compounds in biological systems.
5. Anti-allergy response can be checked through in-silico analysis.
Author Response
Answer to Reviewer’s comment:
- Please discuss the future directions/recommendations and limitations of this study in a separate section
Answer: Thank you for the reviewer’s appropriate comments. As suggested by reviewer, we further discussed the potential use of TA25 as an antiviral therapeutic agent in the final paragraph of the discussion.
- The study already provides insights into the mechanism of action, more detailed studies on how these compounds interact with viral components would be beneficial. For this purpose, in-silico docking, and simulation can be incorporated in this manuscript.
Answer: Thank you for helpful advice. We are trying to identify viral (or host) target molecules in several ways. As suggested by reviewer, in silico docking is a powerful method and we are attempting to identify target molecules and elucidate their mode of action. However, due to technical limitations, it is difficult to include data in this manuscript.
- Studies on the potential development of viral resistance to these compounds would be a practical addition to understand their long-term effectiveness.
Answer: Thank you for helpful advice. We will take this into deep consideration for further research and development of antiviral treatments. However, these studies require more time.
- The inclusion of pharmacokinetic and pharmacodynamic data would provide more comprehensive information about the behavior of these compounds in biological systems.
Answer: Thank you for helpful advice. As suggested by reviewer, we know that pharmacokinetic and pharmacodynamic data are essential for drug development. We will address this during further research and development of antiviral treatments. However, these works require more time.
- Anti-allergy response can be checked through in-silico analysis.
Answer: Thank you for helpful advice. In fact, we have confirmed the anti-inflammatory effects of TA0 and TA25, and are studying the target of host molecules that regulate the immune system. We will further test the anti-allergy response as an independent study.
Reviewer 2 Report
Comments and Suggestions for Authors
Presented for review manuscript provides a short and definitive study of the anti-IAV activities of Tabamide A derivates.
Generally, I find the manuscript well-written, novel, and worth publishing.
Nevertheless, as a Reviewer, I need to point out some. points that in my opinion, could make the manuscript even better.
1. The Tabamide should be described in more detail in the introduction. I think the authors should indicate references carefully. For example, in reference 18, there is no mention of Tabamide A. Please review the references.
2. On panels 2c and 4c, the same scale should be applied. The results seem contradictory at first glance. Please unify the scales.
3. There is no indication of the method used for performing experiments or the calculation of IC50 used to prepare Figure 5. Additionally, indicating the actual results used to prepare the curves is preferable. Forgive me, but too often, I have seen the miscalculated IC50 that does not correspond to data due to the mechanical use of software. Please indicate the data point on the graph or change the chart, and recalculate results if needed.
4. In my strong opinion, the discussion section should be supplemented with pieces of information and a discussion of activities known anti-IAV compounds, especially the doses used in similar experimental setups. For example, a study published https://pubmed.ncbi.nlm.nih.gov/26053018/ shows IC50 for Zanamivir in "in vitro" practical set up in ranges 0.18 - 23 uM. It indicates that the Tabamide A derivates are valid candidates for the antivirals. (I have no connection with the aforementioned publication, and the authors should feel free to select examples.) Please review the literature and briefly discuss active concentrations of Tabamide A and other anti-IAV compounds used in similar assays.
I have one additional comment. I was unable to find the article https://doi.org/10.1016/j.tetlet.2021.153482 in Pubmed. PubMed does not return any results from the "Tabamide" search. The article does not appear in PubMed even if the full title is used for the search. I found the article in the ScienceDirect. The journal claims that is indexed in PubMed. I have no idea if the authors paid for the article. Nevertheless, it is unacceptable. How could anyone cite if the article cannot be found? It is important in times when the Editors and scientists so often discuss the quality of one journal over the others. In that case, the Tetrahedron Letters, from Elsevier, is not a good quality journal. I recommend informing the Editor about the issue.
Author Response
Answer to Reviewer’s comment:
- The Tabamide should be described in more detail in the introduction. I think the authors should indicate references carefully. For example, in reference 18, there is no mention of Tabamide A. Please review the references.
Answer: Tabamides (tabamide A, B, and C), the phenolic amides identified from N. tabacum, are poorly understood for their function, except for two references 20 and 21 that defined these compounds for the first time. Our study showing that these compounds have anti-inflammatory effects is another piece of information (ref. 22). Reference 18 describes several use of N. tabacum as a folk remedy, but not tabamide A.
- On panels 2c and 4c, the same scale should be applied. The results seem contradictory at first glance. Please unify the scales.
Answer: Thank you for the reviewer’s appropriate comments. We changed the format and scale of Figure 4C to that of Figure 2c.
- There is no indication of the method used for performing experiments or the calculation of IC50 used to prepare Figure 5. Additionally, indicating the actual results used to prepare the curves is preferable. Forgive me, but too often, I have seen the miscalculated IC50 that does not correspond to data due to the mechanical use of software. Please indicate the data point on the graph or change the chart, and recalculate results if needed.
Answer: We indicate the experimental method for calculating IC50 in the legend of figure 5 (lines 181-182). We also present the actual results for the curves in Figure 5A.
- In my strong opinion, the discussion section should be supplemented with pieces of information and a discussion of activities known anti-IAV compounds, especially the doses used in similar experimental setups. For example, a study published https://pubmed.ncbi.nlm.nih.gov/26053018/ shows IC50 for Zanamivir in "in vitro" practical set up in ranges 0.18 - 23 uM. It indicates that the Tabamide A derivates are valid candidates for the antivirals. (I have no connection with the aforementioned publication, and the authors should feel free to select examples.) Please review the literature and briefly discuss active concentrations of Tabamide A and other anti-IAV compounds used in similar assays.
Answer: Following the reviewer’s helpful advice, we compared the IC50 values of our compounds with Oseltamivir or Zanamivir in the Discussion section (line 259), and added three references (25-27).
I have one additional comment. I was unable to find the article https://doi.org/10.1016/j.tetlet.2021.153482 in Pubmed. PubMed does not return any results from the "Tabamide" search. The article does not appear in PubMed even if the full title is used for the search. I found the article in the ScienceDirect. The journal claims that is indexed in PubMed. I have no idea if the authors paid for the article. Nevertheless, it is unacceptable. How could anyone cite if the article cannot be found? It is important in times when the Editors and scientists so often discuss the quality of one journal over the others. In that case, the Tetrahedron Letters, from Elsevier, is not a good quality journal. I recommend informing the Editor about the issue.
Answer: Thank to the reviewer’s helpful advice. Although Tetrahedron Letters, published by Elsevier is not listed in MEDLINE, this journal has been widely read by scientists in organic chemistry since 1959. We cited Tetrahedron Letters because our first paper on tabamide A-C is about the newly synthesized organic compounds.
Reviewer 3 Report
Comments and Suggestions for Authors
1. Line 45, please state the full name of vRNP as viral ribonucleoprotein
2. Line 94, formatting
3. Line 158-159, please make sure when discussing the reduction of plaques it is “reduced to” or “reduced by.”
4. Please plot Figure 5 with the actual data point and SD
5. Line 193, please specify the dosage used for TA0 and TA25 in mice
6. Line 274, in the discussion section, please elaborate on how the two antiviral drugs are related to the topic. Efficacy comparison, mechanism comparison, etc., right now, this paragraph is isolated and not related to the manuscript.
7. Line 279, 281, please edit the formatting on baloxavir
Comments on the Quality of English Languageminor edits needed
Author Response
Answer to Reviewer’s comment:
- Line 45, please state the full name of vRNP as viral ribonucleoprotein
Answer: Line 39-40, we are first mentioned ‘vRNP’ full name and abbreviation. Line 45, second use of “vRNP” then we were used abbreviation.
- Line 94, formatting
Answer: Line 94, ‘analysis’ font size was 12, that was changed to 10
- Line 158-159, please make sure when discussing the reduction of plaques it is “reduced to” or “reduced by.”
Answer: Line 159, we are changed “reduced” to “reduced to”
- Please plot Figure 5 with the actual data point and SD
Answer: Thank you for the reviewer’s comments. We are added actual results for the curves in Figure 5A.
- Line 193, please specify the dosage used for TA0 and TA25 in mice
Answer: Thank you for the reviewer’s comments. We added the following to Line 194 regarding the doses of TA0 and TA25 injected into mice, “treated with 20 ml of 10 mM TA0 or TA25” These injection conditions are mentioned in the methods section.
- Line 274, in the discussion section, please elaborate on how the two antiviral drugs are related to the topic. Efficacy comparison, mechanism comparison, etc., right now, this paragraph is isolated and not related to the manuscript.
Answer: Thank you for helpful advice. We added the content indicating that tabamide derivatives can act as RdRp inhibitors to the previous paragraph (Lines 267-283), and linked to the content of next paragraph concerning two antiviral drugs. We also further discussed the potential use of TA25 as an antiviral therapeutic agent in the final paragraph of the discussion.
- Line 279, 281, please edit the formatting on baloxavir
Answer: Line 289, 291, “baloxavir” font size was 12, that changed to 10